# Healthy Eating Index and Nutrition Biomarkers among Army Soldiers and Civilian Control Group Indicate an Intervention Is Necessary to Raise Omega-3 Index and Vitamin D and Improve Diet Quality

**DOI:** 10.3390/nu13010122

**Published:** 2020-12-31

**Authors:** Melissa Rittenhouse, Jonathan Scott, Patricia Deuster

**Affiliations:** 1Consortium for Health and Military Performance, Department of Military and Emergency Medicine, F. Edward Hébert School of Medicine, Uniformed Services University, Bethesda, MD 20814, USA; Jonathan.scott@usuhs.edu (J.S.); patricia.deuster@usuhs.edu (P.D.); 2Henry M. Jackson Foundation for the Advancement of Military Medicine, Bethesda, MD 20817, USA

**Keywords:** diet quality, healthy eating index, military service members, omega-3 index, vitamin D

## Abstract

Diet quality and nutrition status are important for optimal health and military performance. Few studies have simultaneously evaluated diet quality and biochemical markers of nutritional status of military service members. The Healthy Eating Index (HEI) can be used to assess dietary quality and adherence to federal nutrition guidelines. The aim of this study was to assess soldiers’ diet quality and nutritional status and compare results to a civilian control group. Methods: A cross-sectional study was conducted with 531 soldiers. A food frequency questionnaire was used to calculate HEI scores. A blood sample was collected for analysis of select nutrition biochemical markers. Non-parametric analyses were conducted to compare the diet quality and nutritional status of soldiers and controls. Differences in non-normally distributed variables were determined by using the Wilcoxon signed-rank test. Results: Soldiers had an HEI score of 59.9 out of 100, marginally higher than the control group (55.4). Biochemical markers of interest were within normal reference values for soldiers, except for the omega-3 index and vitamin D. Conclusions: This study identified dietary components that need improvement and deficits in biochemical markers among soldiers. Improving diet quality and nutritional status should lead to better health, performance, and readiness of the force.

## 1. Introduction

A healthy mission-ready force is a high priority for the Defense Health Agency (DHA), public health centers, and military leaders. One component of health is nutritional status, which is mentioned often throughout the Department of Defense (DoD), but limited actions have been taken to scientifically and simultaneously evaluate the diet quality and nutritional status of military service members (SMs) by a standardized process. Diet quality varies depending on what foods are available, cultural traditions, and how much time and effort SMs have to seek out high-quality foods. Nutritional status can fall along a continuum from adequate to deficient. In the typical context, nutritional status refers to “an individual’s health condition as it is influenced by the intake and utilization of nutrients” [1]. For example, if an SM is deficient in a particular nutrient, he/she might be unable to perform required tasks. In addition, a biomarker of nutritional status could be abnormal, such as high cholesterol. According to the DHA, the prevalence of obesity in SMs has steadily increased from 5.0% in 1995 [2] to 17.4% [3] in 2018. Obesity rates of males (18.4%) were higher than females (12.6%), and higher in older (28.2%) compared to younger (9.7%) SMs [3]. Optimal diet quality and nutritional status are important for preventing obesity, as well as for overall health and performance.

Federal nutrition guidelines, including the Dietary Guidelines for Americans (DGAs), were developed to help prevent chronic disease in the United States (US) [4]. Every five years, the United States Department of Agriculture (USDA) and Health and Human Services (HHS) publish updated DGA [4,5]. The 2015–2020 DGAs outline how people can improve their overall eating patterns and emphasize the following dietary habits: maintain healthy eating patterns across the lifespan; focus on food variety, nutrient density, and the amount of food; limit calories from added sugar; reduce saturated fat and sodium intake; and shift to healthier food and beverage choices. The DGAs assist health professionals and the public in adopting healthy eating patterns, and are applied when developing policies, programs, and strategies to help prevent chronic diseases. What is clear is that nutrition choices and food habits affect every aspect of life, from physical performance [6,7,8], cognitive performance [9,10,11], sleep [12], and mood [13] to overall health [14]. Thus, a high-quality diet is critical for the health and performance of all soldiers, and assessing a soldier’s adherence to the DGAs would provide pertinent feedback on their dietary choices.

The healthy eating index (HEI) is one standardized process for measuring overall diet quality (e.g., balance, variety, and adequacy), and uses a scoring system to evaluate a set of foods. HEI total scores range from 0 to 100, with 100 indicating complete adherence with the 2015–2020 DGAs. The HEI-2015 overall score is made up of 13 components that reflect different food groups and key recommendations from the 2015–2020 DGAs [15], and it provides an objective assessment of diet quality [16]. Higher HEI scores have been associated with positive health outcomes, including lower risk of cardiovascular disease, diabetes [14], cancer [17,18], and obesity [19]. Lutz et al. reported HEI 2010 scores for military recruits and noted a broad range, with all needing improvement and none scoring as “good quality” [20]. Likewise, Cole et al. [21] evaluated HEI 2010 scores among special operators and found similar results—all were either poor or needed improvement. Together, these results indicate a need to improve the dietary habits of service members.

Importantly, HEI scores have been calculated in many studies, with the largest being the National Health and Nutrition Examination Survey (NHANES). The National Center for Health Statistics continuously conducts the NHANES, which provides timely population-based information on food consumption; their data allow estimates of dietary adequacy and excess to be developed and then used for comparisons [15]. Data from these surveys provide an overview of the total nutrient intake and the current nutritional status across the US. With a diverse force that has various needs, the HEI is currently the only measure available to assess diet quality in the military. It can be used by dietitians to gain a general sense of whether nutrients consumed by SMs are adequate or insufficient [22]. Educating soldiers on areas of dietary improvement is necessary, and strategies for implementing dietary changes might be beneficial across the DoD [8,23,24,25]. As such, we sought to directly compare HEI scores and nutritional status between a cohort of active-duty US Army soldiers and civilians by using age/sex matched NHANES data.

Purpose: The aim of this study was to evaluate Army soldiers’ diet quality and nutritional status by using the HEI-2015 and selected biochemical markers, respectively, and compare the results with a civilian control group of standardized norms from the 2015–2016 NHANES. This study will contribute to our better understanding of diet quality and nutritional status in the military in an effort to improve health, performance, and readiness of the force.

## 2. Materials and Methods

A cross-sectional study was conducted as a collaboration between the Consortium for Health and Military Performance (CHAMP) at the Uniformed Services University (USU) and the US Army’s Comprehensive Soldier and Family Fitness Program (CSF). All subjects gave their informed consent for inclusion before they participated in the study. The protocol was approved by the USU Institutional Review Board (CHAMP-91-2536) on 3 May 2013. Written informed consent was obtained from soldiers prior to data collection. All participants were assigned an ID number during data collection for de-identification purposes.

### 2.1. Participant Recruitment

Soldiers were recruited using flyers, social media, and word of mouth from June 2014 to November 2017 at five US Army installations. The on-site points of contact (POCs) assisted with recruiting and briefing soldiers about the study. Each participant that completed the study, including a fasting blood draw, was compensated with a $25 debit card. Inclusion criteria were being an active-duty soldier, at least 18 years of age, and giving blood following an 8–12 h fast. Data from questionnaires were included for participants who declined the blood draw. USU medical students were excluded.

### 2.2. Dietary Intake

The paper-based version of the 2005 Block food frequency questionnaire (FFQ) (Nutrition Quest, Berkeley, CA) with 110 questions was completed in person to assess dietary eating patterns and calculate HEI-2015 scores. The overall response rate on the FFQ was 93%. Table 1 and Table 2, respectively, show the two HEI component categories, adequacy (nine components) and moderation (four components), with scores being calculated according to set minimum and maximum standards [22]. For the adequacy component, food intake at the level of the set standard or higher receives the maximum number of points, with higher intakes being desirable. For the moderation component, food intake at the level of the set standard or lower receives the maximum number of points, since lower intakes are preferred. Intake scores between the minimum and maximum standards are scored proportionately [16]. Each component is weighted equally at 10 points, and some contain subcomponents (fruit, vegetables, and protein). The total score is calculated from the sum of the individual component scores [16], to a potential maximum score of 100. The HEI uses a density approach, where each component is scored per 1000 kilocalorie [16], to create a standardized approach for comparing diet quality across the general population [16]. The scores of each component indicate dietary patterns, and highlight the components that need improvement. Improving the lowest scoring components will be most beneficial in improving overall diet quality.

### 2.3. Control Group of Normative NHANES Match

The NHANES sample is a noninstitutionalized US civilian population of all ages that reside in all 50 US states and Washington, D.C. The survey uses dietary interviews to examine a nationally representative sample of about 5000 persons each year [26]. A civilian control group, consisting of age/sex standardized norms from the NHANES 2015–2016 survey [26], was compared to HEI scores and biochemical markers of US Army soldiers collected in 2015–2016. The population method was used to calculate the control group (NHANES match) HEI scores. This method calculated the mean intake of dietary constituents, and scoring standards were applied to obtain scores at a group level [27]. The NHANES sample was stratified by age (18–58) and sex, using two-year age increments. Within each stratum (*n* = 23 for each sex), the NHANES sample frame-weighted median (for non-normally distributed variables: HEI, biochemical values) or the mean (for normally distributed variables: demographics) were calculated; these values were then matched to equivalent age/sex strata of soldiers.

### 2.4. Biochemical Markers

Soldiers were required to fast (>8 h) prior to giving a blood sample by arm venipuncture. The biomarkers were not collected to be reflective of intake based on the FFQ, but rather reflective of nutritional status. The biochemical markers were chosen because past research has indicated the lipid, iron, vitamin D, and omega-3 index statuses are possibly low among military service members [8,28,29,30,31,32,33]. The biochemical markers of concern in the military include total cholesterol, low density lipoprotein (LDL), high density lipoprotein (HDL), triglycerides, glucose, hemoglobin (Hgb), hematocrit (HCT), ferritin, iron [33], high sensitivity C-reactive protein (hsCRP), omega-3 index [28,29,32], and serum 25-hydroxy vitamin D (vitamin D) [30,31]. Lipid profiles were analyzed using the NMR lipid profile test at the National Institutes of Health (NIH; Bethesda, MD) National Heart Lung and Blood Institute (NHLBI). Due to changes in the contract, vitamin D was analyzed by NIH and Cleveland Heart Lab, Inc. Vitamin D was analyzed using a DiaSorin chemiluminescence technology immunoassay at the NHLBI and by a Chemiluminescence assay using high-performance liquid chromatography tandem mass spectrometry at Cleveland Heart Lab, Inc. (Cleveland, OH, USA). Hgb and HCT were measured on-site using a handheld Alere HemoPoint H2. Analysis of ferritin was done with the Siemens 2000 immunoassay, and iron was analyzed with the Beckman Coulter DXC600 Pro at Pennington Biomedical Research Center (Baton Rouge, LA, USA). Folate was analyzed by an electrochemiluminescence immunoassay, and hsCRP was analyzed by an immunoturbidimetric assay at Cleveland Heart Lab, Inc. (Cleveland, OH, USA). Glucose level was analyzed on-site from a drop of blood placed on a test strip with the OneTouch Ultra 2 Glucometer (Malvern, PA, USA). For the omega-3 index, 1–2 drops of blood were placed on an antioxidant-treated card on-site, and cards were sent to OmegaQuant (Sioux Falls, SD, USA) for analysis.

### 2.5. Statistical Analysis

Data analyses were completed using IBM SPSS statistics for Windows, Version 25 (Armonk, NY, USA: IBM Corp). Demographic variables were calculated as frequencies. Medians with a 25–75% range were reported for data that were not normally distributed. All comparisons between soldiers and NHANES data were conducted using the Wilcoxon signed-rank test. This test was used because the distributions were non-normal by visually inspecting the distributions of the difference scores between the Army sample and the NHANES control sample. Non-parametric analyses were conducted in the comparison of soldier HEI scores with the control group normative values from the appropriate NHANES strata. Differences in non-normally distributed variables (biochemical markers) were analyzed with effect sizes to show the strength of the comparison. Effect sizes ranging between 0.1 and 0.3 indicated a small difference, 0.3 and 0.5 indicated a moderate difference, and greater than 0.5 indicated a large difference between samples. The sample size varied by data availability, particularly for the biochemical measures. Statistical significance was set a priori at *p* ≤ 0.05 with a Bonferroni correction.

This is an exploratory secondary analysis of a larger parent study. The overall objective of the parent project was to describe the physical, spiritual, and nutritional status of US Army soldiers by using a variety of subjective and objective assessment techniques. Using the Wilcoxon signed-rank test, and with power set at 0.8 and alpha set at 0.05, a sample of 35 would be required to detect a large effect (*r* = 0.5) and a sample of 208 to detect a small effect (*r* = 0.2).

## 3. Results

### 3.1. Participants

A total of 531 soldiers participated in the study, of which there were 382 males, 105 females, and 41 soldiers who did not report their sex information. Soldiers (*n* = 36) were excluded from the dietary analysis because of incomplete dietary data (*n* = 5) or reporting low caloric intake (*n* = 31), which was less than 800 kcal/day for males and 500 kcal/day for females [34,35]. Therefore, the final sample size was 495 participants. Only soldiers with complete data were included in the demographic information (Table 3). Soldiers had an age range of 18–58 years old, with a mean age of 27.

### 3.2. Healthy Eating Index (HEI)

Soldiers (*n* = 492) had a median total score of 59.9 out of 100 on the HEI-2015. Components with the highest scores were refined grains, greens and beans, total protein foods, and whole fruit, whereas components with the lowest scores were whole grains, sodium, and fatty acids. When compared to the control group (NHANES match), soldiers had a slightly higher median HEI score (59.9 ± 10.2) than the NHANES match median HEI score (55.4 ± 3.7) (Table 4). In utilizing the HEI to monitor trends in diet quality over time or assess the effects of interventions, it is necessary to consider what constitutes meaningful differences between groups or change over time. This may also be salient to epidemiologic analyses in interpreting the magnitude of differences in HEI scores that are associated with differences in risk of disease. Previous analyses of HEI scores have shown that the standard deviation of the usual distribution of HEI scores is approximately 10 among children and 11 to 12 among adults (data not shown). Applying an effect size of 0.5 (which may denote a moderate effect), a difference between independent groups of five to six points might be considered meaningful. However, it is possible that the standard deviation may differ by population and assessment tool, suggesting that researchers need to carefully evaluate differences over time or across groups in the context of the range of scores observed in the study and/or similar studies [36].

Figure 1 presents a radar graph depicting the soldiers’ HEI component scores compared to those for the NHANES matched group, according to the HEI-2015. The most notable differences where soldiers surpassed the NHANES control group were for total fruit (3.7 compared with 2.9), whole fruit (4.1 compared with 3.4), greens and beans (4.2 compared with 3.4), refined grains (8.9 compared with 4.2), and added sugar (7.8 compared with 6.1). In contrast, soldiers did more poorly than the NHANES group for whole grains (2.3 compared with 2.9), total protein foods (4.2 compared with 5.0), and seafood and plant proteins (3.7 compared with 4.6).

### 3.3. Biochemical Markers

The medians for the lipid profiles were within the reference ranges for total cholesterol, LDL, HDL, and triglycerides (Table A1). The medians for folate, glucose, and markers of iron status (hemoglobin, hematocrit, ferritin, and serum iron) were also within normal limits. However, the vitamin D median was below normal according to the Endocrine Society and the American Association of Clinical Endocrinologists, with 86.1% of soldiers having levels below 30 ng/mL. More specifically, 97 soldiers had levels of less than 20 ng/mL or less, and 266 soldiers fell between 20–30 ng/mL. Only 127 soldiers had normal (>30 ng/mL) vitamin D levels, with 97 soldiers between 30–39 ng/mL and 30 soldiers at ≥40 ng/mL. When asked about dietary supplement use, 30% of soldiers reported consuming vitamin D supplements, and their median vitamin D levels were significantly higher than those not taking supplements.

The omega-3 index was assessed among soldiers and revealed an average score of 3.8%, which is well below the recommended level of >8% [27]. No solider presented with an omega-3 index >8%. Only 16% of soldiers reported taking a supplement containing omega-3 fatty acids, indicating that blood levels are primarily reflective of dietary intake. Of those who reported taking a supplement, their average omega-3 index was significantly higher (4.5%) than those not taking a supplement. Because the timing, compliance, and dosing of supplements were not detailed, other details cannot be provided.

The biological marker levels of soldiers were compared to the control group (NHANES match) for total cholesterol, LDL, HDL, triglycerides, glucose, Hgb, HCT, ferritin, iron, and hsCRP. The omega-3 index or vitamin D data were not available for the control group (NHANES match). All biochemical markers except for cholesterol, ferritin, and CRP were significantly (*p* ≤ 0.5) different from the control group (NHANES match). The largest difference between groups was the level of HDL in males, which had a moderate effect size.

## 4. Discussion

Previous research among soldiers has demonstrated less than desirable eating behaviors; in particular, they fall short of the recommended intake of fruit, vegetables, and whole grains, while exceeding recommendations for total and saturated fats, refined grains, and added sugars [11,37,38,39].

To that end, the HEI is of interest, and has been used in several studies of military personnel—all show low adherence to the DGAs [8,11,20,21]. Overall, the present study shows that the dietary habits of our SMs are not consistent with the DGAs, and are likely to put them at risk for multiple chronic diseases in the future. Importantly, coupled with less than adequate diets, their nutritional profiles indicated low vitamin D and omega-3 levels of concern. Across the DoD, we need to promote and support healthy dietary habits so our SMs can meet the DGAs and improve their nutritional status.

### 4.1. Healthy Eating Index

The protective effects of fruit and vegetable consumption from non-communicable diseases have been indicated in multiple studies, yet many Americans fail to meet standards recommended by the DGAs [4,15,40,41]. In addition, excess caloric energy and high saturated fat and refined grain intake can lead to weight gain, increased adiposity, chronic disease, and adverse health consequences [14,19,42,43]. In contrast, a high-quality diet across adulthood has been associated with better physical performance in older age [42]. The HEI is an important index of diet quality and, to our knowledge, this is the first study to comprehensively evaluate the diet quality of US Army soldiers and compare the results to an age/sex matched civilian sample from the NHANES. We found specific dietary components that should be improved, as the soldiers’ overall HEI score indicated low adherence to the DGAs. In particular, our results align with previous research indicating that soldiers commonly fall short on consuming the recommended amount of vegetables and whole grains, and exceed the recommendations for saturated fat and sodium [11,37,38,39].

In comparison to the NHANES matched control group, soldiers’ scores were slightly higher on several HEI components, even though both groups had a total HEI score of 55–60 out of 100. Data from both groups are consistent with previous research indicating that the total HEI-2015 score for US civilians was 59 out of 100 [44]. Therefore, neither civilian nor SM dietary patterns align well to the DGAs, which clearly indicates suboptimal diet quality.

Specific improvements soldiers and civilians can make in their diet include consuming more whole grains, monitoring sodium (unless soldiers have increased needs based on training level or current mission location), and improving their intake of fatty acids by incorporating more monounsaturated and polyunsaturated fats and reducing saturated fat. Over one-third of the soldiers (*n* = 235) in this study were 17–26 years old, which is important to note when considering ways to improve one’s health, as well as for comparison of the HEI scores to future and previous studies with a larger age range.

### 4.2. Biochemical Markers

The results pertaining to biochemical markers indicate that soldiers and the age/sex matched NHANES group would be considered nutritionally adequate. Although statistically significant differences between the two groups were noted, none were clinically significant, since the medians for all markers were within reference ranges. However, the biomarkers for vitamin D and the omega-3 index were strikingly low—approximately 86.1% of soldiers had vitamin D levels below the cutoff of 30 ng/mL. With regard to omega-3 status, about 61% (*n* = 284) had an omega-3 index considered high risk (<4%), and 39% (*n* = 185) had an intermediate risk (4–8%), while none had a low risk (>8%). Current research shows that the average American is only around 4% [26,27], and this cohort of soldiers is slightly lower. Harris et al. [27,32,45] provided these ranges to stratify for cardiovascular risk. Our data are somewhat consistent with those of Anzalone et al. [46], who found that 34% and 66% of athletes had an omega-3 index in the high and intermediate risk ranges, respectively. In summary, the diet quality and select biomarkers of nutritional status need to be improved in SMs. More research is needed to understand how to best get SMs to improve their diet quality, and thereby enhance overall health, performance, and readiness of the force.

### 4.3. DoD Programs to Consider

This study identified specific components for improving diet quality and deficits in biochemical markers that impact nutritional status. Education and service programs can be used to address the necessary dietary adjustments and deficits highlighted in this study. One such program is the Go for Green^®^ joint service (Army, Air Force, and Navy) program that uses a stoplight color-coding system to help SMs improve food choices in dining facilities [21]. Other programs include the Air Force Food Transformation Initiative, which is designed to provide airmen greater variety, availability, and quality of food, while maintaining home station and warfighting feeding capabilities. Another, the Navy Operation Fitness and Fueling (NOFFS), provides physical fitness and nutrition information to sailors. In addition, Fuel to Fight^®^ helps marines make healthy choices that provide optimal fueling by using a stoplight color-coding system similar to Go for Green^®^. Education and behavior modifications will be key in obtaining optimal nutritional status and a mission-ready force.

### 4.4. Strengths and Limitations

The strengths of this study include being the first to obtain dietary quality and nutritional status for Army soldiers and compare them with age and sex matched US civilians. The results provide the foundation for evaluating diet quality and nutritional status over time, and can be used to improve the overall health and readiness of the force.

One limitation of this study is the two different methods, the block FFQ and a 24 h dietary recall, used to capture dietary intake. However, previous research has demonstrated that FFQs provided comparable results to 24 h dietary recalls [47,48]. Both methods also used the Food and Nutrient Database for Dietary Studies (FNDSS) to retrieve nutrient values, and with minimal changes from year to year, we expect similar results allowing for comparison [47,48]. Methods for questionnaire development and validation were previously reported [47].

Another limitation of the study was that, given the predominately Caucasian demographics, recommendations on dietary quality and nutritional status by race were not possible for the HEI scores or biochemical markers. However, these limitations do not take away the clear message: we need to improve the diet quality our SMs consume to protect their health and optimize their performance and readiness.

## 5. Conclusions

This cross-sectional study identified components of diet quality by using HEI scores that can be improved; it also revealed deficiencies in vitamin D and the omega-3 index from biochemical marker analysis in a cohort of Army soldiers. These data were compared to an age/sex matched group from the NHANES to evaluate differences in diet quality and nutritional status between military and civilian populations. The results revealed that many improvements are needed for both populations, and identified specific components of the soldiers’ health that needed improvement. Diet quality and nutritional status directly impact overall performance and mission readiness of the force.

## Figures and Tables

**Figure 1 nutrients-13-00122-f001:**
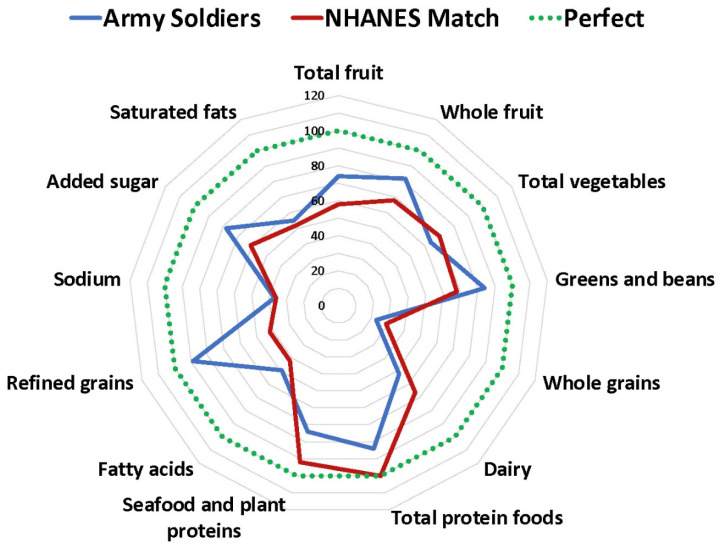
Soldiers’ HEI component scores compared to the NHANES matched group, according to the HEI-2015.

**Table 1 nutrients-13-00122-t001:** Adequate intake HEI component scoring standards.

HEI Components	Standard (Per 1000 Kcal)	Max Score
Total fruit	≥0.8 cup equivalents	5
Whole fruit	≥0.4 cup equivalents	5
Total vegetables	≥1.1 cup equivalents	5
Greens and beans	≥0.2 cup equivalents	5
Whole grains	≥1.5 oz equivalents	10
Dairy	1.3 cup equivalents	10
Total protein foods	≥2.5 oz equivalents	5
Seafood and plant protein	≥0.8 oz equivalents	5
Fatty acids	(MUFA + PUFA)/SFA *	10

* MUFA: monounsaturated fatty acid; PUFA: polyunsaturated fatty acid; SFA: saturated fatty acid.

**Table 2 nutrients-13-00122-t002:** Moderation intake HEI components and standards for maximum and minimum scores.

HEI Components	Standard for Maximum Score (10)	Standard for Minimum Score (0)
Refined grains	1.8 oz equivalents (per 1000 kcal)	≥4.3 oz equivalents (per 1000 kcal)
Sodium	≤1.1 g (per 1000 kcal)	≥2.0 g (per 1000 kcal)
Added sugars	≤6.5% of total energy	≥26% of total energy
Saturated fats	≤8% of total energy	16% of total energy

**Table 3 nutrients-13-00122-t003:** Demographics of the US Army soldier cohort (*n* = 531).

Category	# of Participants (% of Category)
Sex	*n* = 487
Female	105 (22)
Male	382 (78)
*No response*	41
Race	*n* = 415
Asian	16 (3.9)
Black or African American	99 (23.9)
Native American/Alaskan Native	4 (0.9)
Native Hawaiian/Pacific Islander	6 (1.4)
White or Caucasian	248 (59.8)
Other	21 (5.0)
*No response*	134
Rank	*n* = 431
Enlisted	391 (91)
Officer	40 (9)
*No response*	97
Age	*n* = 425
25 and under	217 (51)
26 and older	208 (49)
*No response*	103

*n* = number of participants, # = number, and (%) is percentage of the category. *No response* indicates participants with incomplete datasets.

**Table 4 nutrients-13-00122-t004:** Comparison of median HEI scores for Army soldiers and NHANES match for civilians.

HEI Component	Army Soldier Median (25–75% Range)	NHANES Match Median (25–75% Range)
Adequate components		
Total fruit	3.7 (2.4–5.0)	2.9 (2.4–3.0)
Whole fruit	4.1 (2.0–5.0)	3.4 (2.9–3.9)
Total vegetables	3.2 (2.4–4.3)	3.5 (3.1–4.0)
Greens and beans	4.2 (2.3–5.0)	3.4 (2.3–3.9)
Whole grains	2.3 (1.4–4.1)	2.9 (2.4–3.4)
Dairy	5.2 (3.7–7.4)	6.6 (5.9–7.6)
Total protein foods	4.2 (3.5–5.0)	5 (5–5)
Seafood and plant proteins	3.7 (1.8–5.0)	4.6 (4.0–4.8)
Fatty acids	4.9 (3.3–6.5)	4.2 (4.2–4.2)
Moderation components		
Refined grains	8.9 (7.2–10)	4.2 (3.1–5.2)
Sodium	3.7 (2.0–5.7)	3.6 (3.6–3.6)
Added sugar	7.8 (5.6–9.3)	6.1 (5.5–6.9)
Saturated fats	5.5 (3.6–7.3)	5.2 (5.2–5.2)
Total HEI	59.9 (53–67)	55.4 (52–58)

## Data Availability

Data is available upon reasonable request.

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
