# Peer review of "Healthy Eating Index and Nutrition Biomarkers among Army Soldiers and Civilian Control Group Indicate an Intervention Is Necessary to Raise Omega-3 Index and Vitamin D and Improve Diet Quality"

_nutrients, 2020, doi:10.3390/nu13010122_

Round 1

Reviewer 1 Report

Thank you for the opportunity to review this article. The work is very interesting, but some aspects should be taken into account before publication.

Comments and suggestions for Authors:

First of all authors should once more review the guidelines for authors. Please correct the references. Keywords: It  needs to be sorted in alphabetical order. The text is unjustified. The font should also be uniform throughout the manuscript (including tables).

The appearance and readability of tables (e.g. table 3) are unacceptable. Please correct the layout.

There is no information about obtaining the consent of the Bioethics Committee. Have the authors obtained such consent, if so please provide the number.

Introduction:

  • Authors should include a literature review in the introduction.
  • Authors should improve the introduction including the latest articles published for example in the MDPI platform or others, about other research in this field
  • Authors should better justify the study, highlighting clearly the gap in the current knowledge. What is the novelty of the study?

Materials and Methods

  • Line 100 – please add information about the sample size. What were the group inclusion and exclusion criteria? Please discuss power calculation and how the sample size is adequate.

Result:

  • Line 175 – why 41 soldiers did not give information about their sex?
  • Figure 1 - The caption of the figure should be placed under the figure

Reviewer 2 Report

The submitted article presents a very original and relevant theme, since the adequate nutritional status of military service members is crucial for the performance of their professional activity, consequently for the security of the country.
In general, the article is well written, however I have some suggestions for improvement in the Materials and Methods section:

1) The inclusion and exclusion criteria in the study must be clearly described;

2) It must be indicated how the FFQ was applied, which may help to understand the high percentage of lack of information in relation to some parameters. This percentage should be clearly identified;

3) What is the criterion used to choose biomarkers? Because there was no dosage of vitamins, especially vitamin C, which is the best biomarker of fruit and vegetable intake, or the urine test to assess the actual sodium intake. The total nitrogen intake was also not evaluated. So why has plasma albumin not been evaluated, since it is an excellent indicator of nutritional status? I think this part could be improved in an upcoming study. I believe that the choice of biomarkers of nutritional status was not well done;

4) The FFQ has not been validated for the population studied and therefore it would have been more appropriate to have assayed some biomarkers of food intake (eg protein, vitamin C, sodium, etc.) that somehow give greater reliability to the subjective results the food intake assessment questionnaires;

5) The methods used to measure the biomarkers are not described or referenced;

6) Regarding the statistical analysis, I believe that it is poorly described. Why did they define that all biochemical variables had non-normal distribution? What was the test applied to assess the distribution of variables? What non-parametric analyzes are used (line 165)? Why was no test applied to assess the possible presence of differences in frequency distribution by sex, race, rank and age? It must be done and added to Table 1;

In the results:
The values ​​presented in table 5A for “CRP” have not been described. The median values ​​were higher than the recommended value for both the “soldiers” group and the “NHANES match” group. C-reactive protein is a general marker of the inflammatory process, values ​​higher than those recommended indicate the presence of some type of inflammatory process. This aspect should be explored in the results and correlated with the values ​​obtained for the other biomarkers and with the results of the fatty acids. For that, a Spearman correlation or a univariate or multivariate linear / logistic regression must be performed.

In the discussion:
Line 304 - Bibliographic references must be included.

Reviewer 3 Report

A very interesting and interesting job. I fully accept for printing

Author Response

No further explanation was necessary for this reviewer. 

Round 2

Reviewer 1 Report

Thank you for the opportunity to review this resubmission.  Authors have done a nice job addressing reviewers' comments. Thank you. I am ok with acceptance.